# Preparation, Characterization of ZnTiO_3_/ZnO Composite Materials and Their Photocatalytic Performance

**DOI:** 10.3390/nano12081345

**Published:** 2022-04-14

**Authors:** Mao Tang, Shiji Lu, Lili He, Xiaodong Zhu, Wei Feng, Wanming Zhang

**Affiliations:** 1School of Mechanical Engineering, Chengdu University, Chengdu 610106, China; tangmao@cdu.edu.cn (M.T.); l3221668248@163.com (S.L.); jasmine95762021@163.com (L.H.); fengwei233@126.com (W.F.); 2Sichuan Province Engineering Technology Research Center of Powder Metallurgy, Chengdu 610106, China; 3School of Resources and Environment, Xichang University, Xichang 615000, China

**Keywords:** ZnO, ZnTiO_3_/ZnO, photocatalytic performance, sol–gel

## Abstract

With zinc acetate and butyl titanate as raw materials, pure ZnO and ZnTiO_3_/ZnO composite photocatalysts were synthesized by a sol–gel method and calcined at 550 °C. The crystal structure, morphology, surface area, optical property, and element valence states of samples were characterized and the photocatalytic activity of the prepared photocatalysts were assessed by the degradation of rhodamine B. Results show that the crystal structure of ZnO is a hexagonal wurtzite phase with a band gap of 3.20 eV. When the Zn/Ti molar ratio reaches 0.2, ZnTiO_3_ phase appears and ZnTiO_3_/ZnO composite forms, which advances the transfer of photogenerated charges. ZnTiO_3_/ZnO (Ti/Zn = 0.2) exhibits the highest photocatalytic activity, and the degradation degree of RhB reaches 99% after 60 min, which is higher than that of pure ZnO (90%). An exorbitant Ti/Zn molar ratio will reduce the crystallinity and form more amorphous components, which is not conducive to photocatalytic performance. Therefore, when the Ti/Zn molar ratio exceeds 0.2, the photocatalytic activities of ZnTiO_3_/ZnO composites decrease.

## 1. Introduction

Environmental pollution is a major issue related to the sustainable development of society and economy. In recent years, the use of photocatalytic technology to degrade sewage has been rapidly developed [1,2,3]. Among many photocatalytic materials, ZnO has attracted extensive research attention due to its low cost, good chemical stability, and high electron mobility [4,5,6]. Pure ZnO will encounter some problems when it is directly applied to photodegradation. The forbidden bandwidth of ZnO is large (Eg = 3.37 eV), which can only absorb a small amount of ultraviolet rays in the sunlight. On the other hand, the photogenerated electrons generated by excitation are apt to reunite with holes [7,8]. It is necessary to modify ZnO to improve its photocatalytic activity by ion doping [7,9,10], precious metal deposition [11,12], semiconductor compounding [13], and so on.

Due to the different positions of the conduction band and the valence band, the photo-generated charges can quickly migrate between the two-phase interfaces, reducing the recombination probability of photogenerated electrons and holes, and prolonging the carrier lifetime. Therefore, combining ZnO with other semiconductors to form heterojunctions is helpful to generate more free radicals and improve the photodegradation effect [14]. The most studied ZnO semiconductor composite systems are TiO_2_/ZnO [1,15], ZnS/ZnO [16], Cu_2_O/ZnO [17,18], Fe_2_O_3_/ZnO [19], and CdS/ZnO [20]. He et al. [17] prepared Cu_2_O/ZnO composite photocatalyst and found that—compared with pure ZnO—the visible light absorption was increased, the photogenerated charge separation was improved, and its photocatalytic activity was higher than that of pure ZnO and pure Cu_2_O.

The present work adopts the sol–gel method, with zinc acetate as the precursor, and pure ZnO and ZnTiO_3_/ZnO composite materials with different Ti/Zn molar ratios were obtained at a calcination temperature of 550 °C. The crystal structure, morphology, surface area, optical property and element composition of the obtained photocatalysts were characterized, and their photocatalytic performance was studied by assessing the degradation degree of rhodamine B (RhB).

## 2. Experimental Section

### 2.1. Material Preparation

Zinc acetate (Analytical Reagent, AR), butyl titanate (AR), anhydrous ethanol (AR), and rhodamine B (AR) were purchased from Chengdu Chron Chemicals Co., Ltd. (Chengdu city, Sichuan province, China). All the chemical reagents were used directly without further purification.

An appropriate amount of zinc acetate was dissolved in a 5:3 volume ratio of anhydrous ethanol and deionized water mixed solution, placed in a 70 °C water bath, stirred for 2 h to obtain a gel, and then aged for 48 h to obtain a dry gel. After milling, it was heat-treated at 550 °C for 1 h to obtain pure ZnO powder.

While adding zinc acetate and different contents of butyl titanate, the remaining steps are the kept same to prepare the ZnTiO_3_/ZnO composite photocatalysts. In this study, the molar ratios of Ti/Zn were 0.2, 0.4, 0.6, 0.8, and 1.0. ZnTiO_3_/ZnO composite materials with different Ti/Zn molar ratios were marked as TZ-0.2, TZ-0.4, TZ-0.6, TZ-0.8, and TZ-1, respectively.

### 2.2. Characterization

The crystal structure of the photocatalysts was analyzed by X-ray diffraction (DX-2700 X-ray diffractometer, Dandong Haoyuan Instrument Co. Ltd., Dandong, China, XRD). The test current was 30 mA, the voltage was 40 kV, the scanning angle was 20–70°, and the scanning speed was 0.06°/s. The morphology (SEM and TEM) was observed using a FEI-Inspect F50 scanning electron microscope and a FEI-Tecnai G2-F20 transmission electron microscope (FEI Company, Hillsboro, OR, USA). The BET specific surface area was measured by a Mike ASAP2460 analyzer (Mike Instrument Company, Atlanta, GA, USA). The recombination of photo-generated charges was measured by a F-4600 fluorescence spectrometer with an excited wavelength of 300 nm (F-4600, Shimadzu Group Company, Kyoto, Japan, PL). The optical absorption was analyzed by a UV-3600 UV–vis spectrophotometer (UV-3600, Shimadzu Group Company, Kyoto, Japan, DRS). The element composition and valence state were measured by a Thermo Scientific K-Alpha multifunctional surface analysis system (Thermo Scientific K-Alpha, Kratos Ltd., Manchester, Britain, XPS).

### 2.3. Photocatalysis Experiment 

Taking 100 mL (10 mg/L) RhB aqueous solution as the target pollutant, 0.1 g of sample was added into a beaker containing 100 mL RhB aqueous solution. The mixture was stirred for 30 min to achieve adsorption and desorption equilibrium. A 250 W xenon lamp was used as the UV–vis light source (300–800 nm). Samples were taken every 15 min and the absorbance (A) was measured at λ = 553 nm. The formula (A_0_ − A_t_)/A_0_ × 100% was adopted to calculate the degradation degree, where A_0_ and A_t_ are the initial absorbance and the absorbance at time t, respectively.

## 3. Results and Discussion

### 3.1. XRD Analysis

Figure 1 gives the XRD patterns of samples. The diffraction peaks in the XRD pattern of pure ZnO appearing at 31.8°, 34.4°, 36.3°, 47.6°, 56.6°, 62.9°, 66.4°, 67.9° are indexed to the ZnO hexagonal phase, with corresponding reflections (100), (002), (101), (102), (110), (103), (200), (112), indicating that ZnO shows a hexagonal wurtzite structure [21]. Pure ZnO exhibits high diffraction peak intensity and good crystallinity. In the patterns of ZnTiO_3_/ZnO samples, in addition to the peaks of ZnO, the diffraction peaks at 32.8°, 35.3°, 40.5°, 48.9°, 53.4°, 61.8°, 63.4° are indexed to the ZnTiO_3_ phase, with corresponding reflections at (104), (110), (113), (024), (116), (214), (300). As the Ti/Zn molar ratio increases, the peak intensity of ZnTiO_3_ gradually increases, indicating that a ZnTiO_3_/ZnO composite materials form in the samples. It is worth noting that with the increasing Ti/Zn molar ratio, the peak intensity of ZnO is significantly reduced, indicating that the crystallinity declines and the amorphous component gradually increases [22,23,24]. The average grain size can be calculated by the Sherrer formula as follows: D = 0.89 λ/βcosθ. Where D, λ, β, and θ are the average crystalline size (nm), wavelength of the X-ray (CuKα = 1.54056 Å), full width at half maximum (fwhm) intensity of the peak, and the diffraction angle, respectively. The grain sizes of pure ZnO, TZ-0.2, TZ-0.4, TZ-0.6, and TZ-0.8 are 36.2, 24.1, 28.3, 24.0, and 27.8 nm, respectively [25]. Particularly, the diffraction peak intensity of TZ-1 is very weak. It is difficult to distinguish specific diffraction peaks, implying that the amorphous component further increases.

### 3.2. SEM and TEM Analyses 

Figure 2 gives the SEM images of pure ZnO (Figure 2a,b) and TZ-0.2 (Figure 2c,d). It can be seen that the pure ZnO is granular, and the size of a single particle is between 40 and 100 nm. The particle morphologies of TZ-0.2 were irregular and the size distribution is broad. Figure 2e–h shows the element distribution of TZ-0.2. The sample contains three elements (Ti, Zn, O), which are basically uniformly distributed in the matrix.

TEM and HRTEM images of pure ZnO (Figure 3a,b) and TZ-0.2 (Figure 3c,d) are shown Figure 3. The pure ZnO particles are relatively dispersed; however, the average size is significantly higher than that of TZ-0.2. The interplanar spacing in Figure 3b is 0.25 nm, which corresponds to the (101) plane of the ZnO hexagonal phase [26]. In Figure 3d, the marked interplanar spacing of 0.28 nm corresponds to the (100) plane of the ZnO hexagonal phase [27].

### 3.3. BET Analysis

It is observed from the TEM images that the particle size of TZ-0.2 is smaller than that of pure ZnO, and the morphology has changed significantly, thus their specific surface areas may be quite different. Figure 4 presents the nitrogen adsorption–desorption isotherms and pore size distribution curves of pure ZnO (Figure 4a) and TZ-0.2 (Figure 4b). The pore size of TZ-0.2 is smaller than that of pure ZnO, and is concentrated within 10 nm. The specific surface area of pure ZnO is 23.5 m²/g, and it is 67.1 m²/g for TZ-0.2. Compared with pure ZnO, the grain size of TZ-0.2 decreases obviously, which favors increasing the specific surface area. The larger surface area is beneficial to improving the photocatalytic activity as there are more active sites [20,28].

### 3.4. Optical Property

When the photogenerated electrons are excited to the conduction band, leaving holes in the valence band, they are easy to return to the valence band and recombine with the holes. Simultaneously, the photon will be released, which constitutes the PL peak. Therefore, the lower the PL peak intensity, the less recombination of photogenerated electrons and holes [29,30,31]. Figure 5 shows the PL spectra of samples. The PL peak intensity of ZnTiO_3_/ZnO composites varies greatly with the molar ratio of Ti/Zn. The peak intensity of TZ-0.2 is lower than that of pure ZnO, indicating that the recombination of photogenerated charges in TZ-0.2 is lower than that of pure ZnO, and exhibits higher quantum utilization. When the heterojunctions are formed between the two semiconductors, it is beneficial to the migration of photogenerated charges at the phase interfaces, which inhibits the recombination with a lower PL peak intensity [14,32]. Unexpectedly, when the Ti/Zn mole ratio reaches 0.4, the PL peak intensity of ZnTiO_3_/ZnO composites is higher than that of pure ZnO, showing lower separation of photoinduced charges. It is generally believed that appropriate defects are beneficial to capturing photogenerated charges and inhibiting the recombination. However, excessive defects will introduce new recombination centers, which is not conducive to improving quantum efficiency [26,33]. According to the XRD results, with the increasing Ti/Zn molar ratio, the diffraction peak intensity of ZnTiO_3_/ZnO composite decreases and the peak width increases, implying that the crystallinity decreases and the amorphous composition increases, which leads to the formation of excessive defects. Consequently, the separation of photogenerated charges decreases [34,35].

The UV–vis absorption spectra (a) and band gap (b) of pure ZnO and TZ-0.2 in the wavelength range of 200–800 nm is shown in Figure 6. Pure ZnO shows strong absorption in ultraviolet region. However, the absorption declines sharply in visible region. The band gap of pure ZnO is determined to be 3.20 eV. Compared with pure ZnO, the absorption edge of TZ-0.2 does not move appreciably, which also exhibits a band gap of 3.20 eV. It is worth noting that the absorptions of TZ-0.2 in both ultraviolet and visible regions increase slightly.

### 3.5. Element Composition and State

To examine the chemical states and surface compositions of ZnO and ZnTiO_3_/ZnO composite, XPS testing was performed and the results are shown in Figure 7. Take TZ-0.2 as an example to analyze the element composition and chemical state. The overall survey spectra indicated that TZ-0.2 contains Zn, O, C, and Ti elements. Figure 7b is the spectrum of O 1s, with two peaks appearing at 530.2 eV and 531.4 eV, corresponding to lattice oxygen and surface hydroxyl groups [36,37]. Figure 7c shows the spectrum of Zn 2p, which can be decomposed into two peaks at 1021.3 eV and 1044.3 eV, corresponding to Zn 2p_3/2_ and Zn 2p_1/2_, implying that Zn exists in the form of Zn^2+^ [27,36]. Figure 7d shows the spectrum of Ti 2p. The peaks at 458.5 eV and 463.6 eV are attributed to Ti 2p_3/2_ and Ti 2p_1/2_, indicating that Ti in the sample is in a +4 state [38].

Figure 7e shows the total spectra of pure ZnO and ZnTiO_3_/ZnO composites. The binding energies are listed in Table 1. Compared to pure ZnO, the binding energies of ZnTiO_3_/ZnO composites shift, which is due to the interaction of Ti element with Zn and O elements [39,40].

The element compositions, Ti/Zn molar ratios, as well as [O]/([Ti] + [Zn]) molar ratios are summarized in Table 2. The results show that there is a certain deviation between the element composition obtained by XPS results and the theoretical values. XPS is a typical surface analysis method for qualitative and semi quantitative analysis. The test area is generally the sample information with a size of hundreds or even tens of microns on the sample surface and the depth of 1–10 nm, which does not represent the overall properties of the sample. However, with the increase in the theoretical value of Ti/Zn molar ratio, the Ti/Zn molar ratio of XPS results also shows an increasing trend.

### 3.6. Photocatalytic Activity

Figure 8 shows the degradation curves of samples against rhodamine B. After 60 min, the degradation degrees of pure ZnO and TZ-0.2 are 90% and 99%, respectively. With the increase in Ti/Zn mole ratio, the degradation degree first increases and then decreases. When the molar ratio of Zn/Ti reaches 0.2, a new phase ZnTiO_3_ appears and a ZnTiO_3_/ZnO semiconductor composite structure forms. The migration of photogenerated charges between semiconductor interfaces inhibits the recombination, which is beneficial to the separation of photogenerated charges [16,20]. The photodegradation results are consistent with the PL spectra. It must be mentioned that a high Ti/Zn molar ratio is not conducive to photocatalytic performance. When the Ti/Zn molar ratio exceeds 0.2, the degradation degrees of ZnTiO_3_/ZnO composites are less than pure ZnO. XRD analysis shows that the intensity of ZnO diffraction peaks decreases significantly with the rising Ti/Zn molar ratio, implying that the crystallinity decreases and the amorphous composition increases, which is not conducive to photocatalytic performance [3,41].

The first-order kinetics curves of photodegradation of RhB are shown in Figure 9. Ln (C_t_/C_0_) shows a linear relationship with the reaction time t, and the reaction rate constant k can be calculated by the formula kt = −ln(C_t_/C_0_) (t is the reaction time, C_t_ represents the concentration of RhB at time t, and C_0_ represents the initial concentration). The k value of pure ZnO is 0.033 min^−1^, and the k value of TZ-0.2 is the highest, reaching 0.046 min^−1^, which exhibits the fastest reaction rate.

## 4. Conclusions

Pure ZnO and ZnTiO_3_/ZnO composite photocatalysts were prepared by sol–gel method. The crystal structure, surface morphology, specific surface area, optical properties, elemental composition, and state were analyzed. ZnO exhibits hexagonal wurtzite structure after calcination at 550 °C. With the increase in the Ti/Zn molar ratio, the grain size of ZnO decreases. When the Zn/Ti molar ratio reaches 0.2, a new phase ZnTiO_3_ appears and ZnTiO_3_/ZnO semiconductor composites form. ZnTiO_3_/ZnO(Ti/Zn = 0.2) shows the best photocatalytic performance. After 60 min of xenon lamp irradiation, the degradation degree of RhB reaches 99%, and the k value reaches 0.046 min^−1^, which is higher than 90% and 0.033 min^−1^ of pure ZnO. When the Ti/Zn molar ratio exceeds 0.2, the degradation degrees of ZnTiO_3_/ZnO composites decrease, which can be attributed to the reduced crystallinity at a high Ti/Zn molar ratio, generating more amorphous components, forming the recombination centers, and reducing the separation of photogenerated charges.

## Figures and Tables

**Figure 1 nanomaterials-12-01345-f001:**
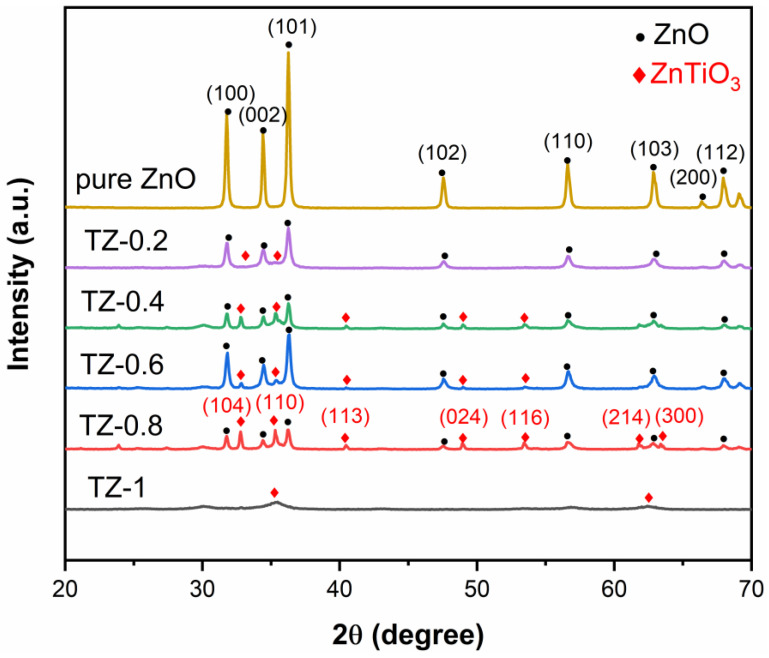
XRD patterns of samples.

**Figure 2 nanomaterials-12-01345-f002:**
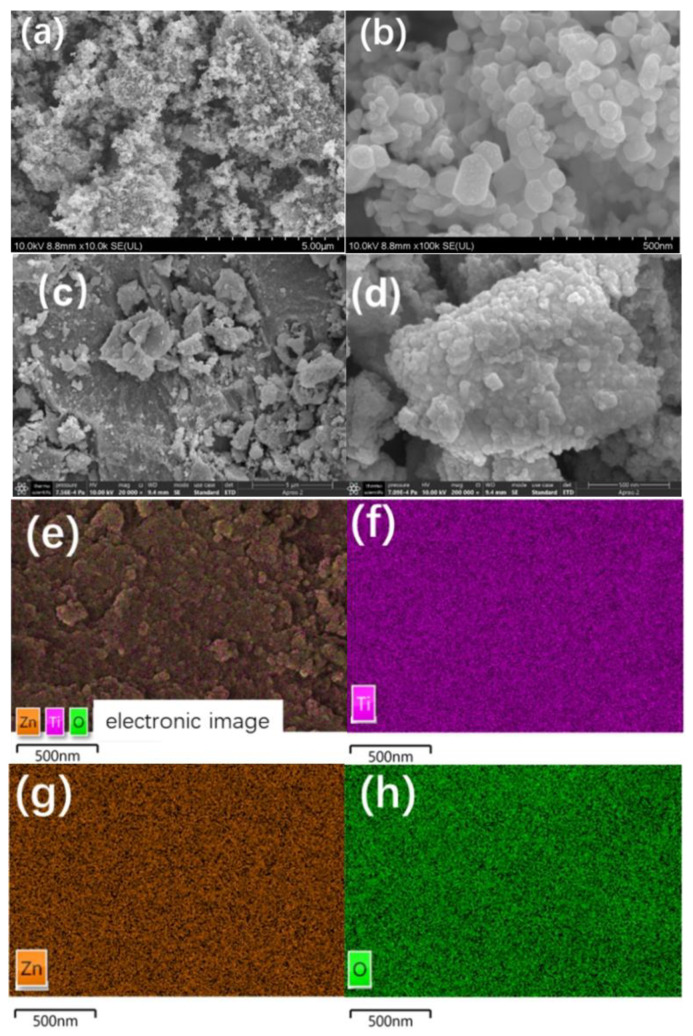
SEM images of pure ZnO (**a**,**b**), TZ-0.2 (**c**,**d**), and elemental mapping images of TZ-0.2 (**e**–**h**).

**Figure 3 nanomaterials-12-01345-f003:**
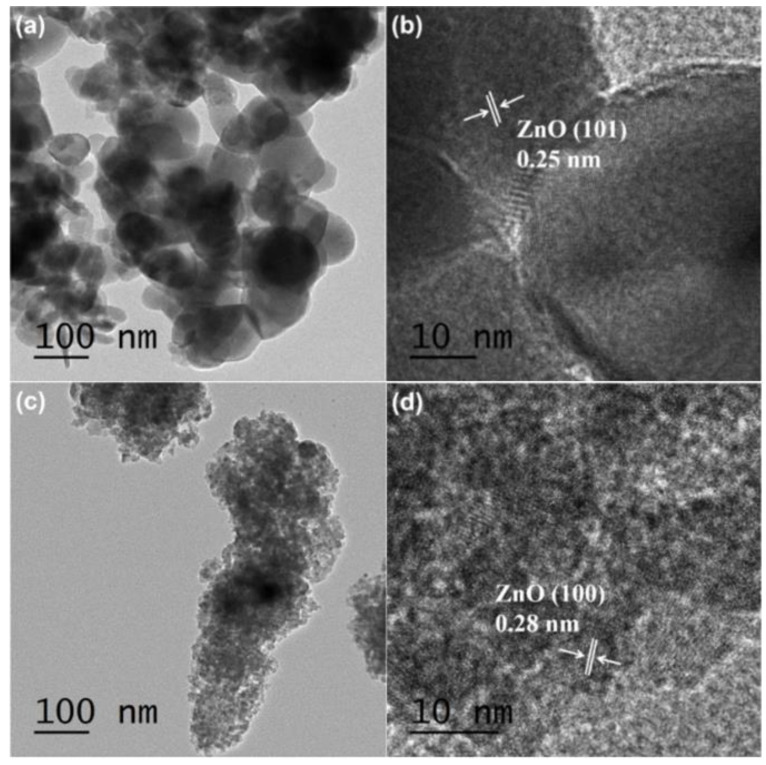
TEM and HRTEM images of pure ZnO (**a**,**b**) and TZ-0.2 (**c**,**d**).

**Figure 4 nanomaterials-12-01345-f004:**
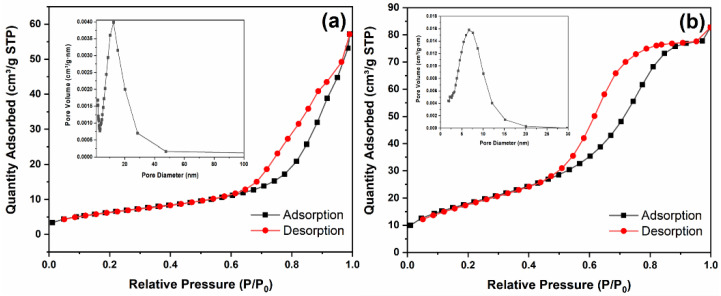
Nitrogen adsorption–desorption isotherms and pore size distribution curves of samples: (**a**) pure ZnO, (**b**) TZ-0.2.

**Figure 5 nanomaterials-12-01345-f005:**
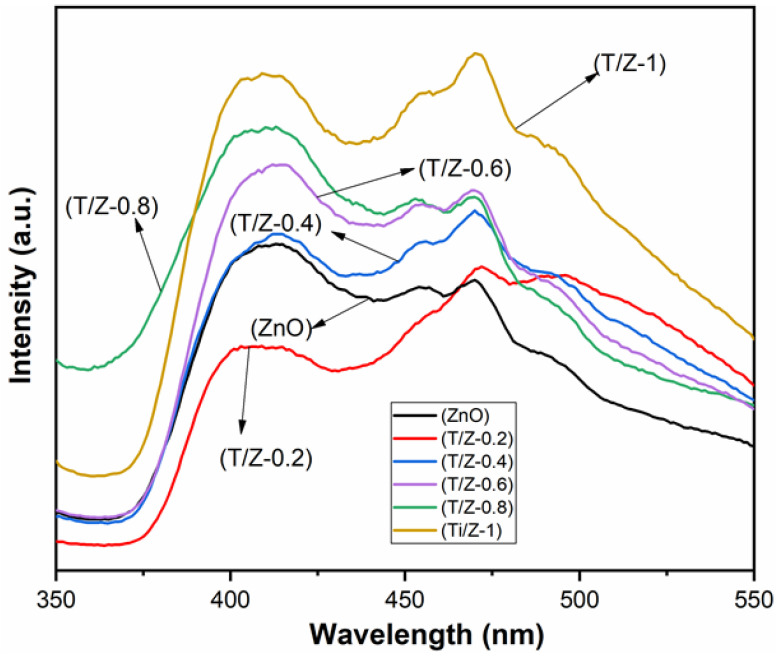
PL spectra of samples.

**Figure 6 nanomaterials-12-01345-f006:**
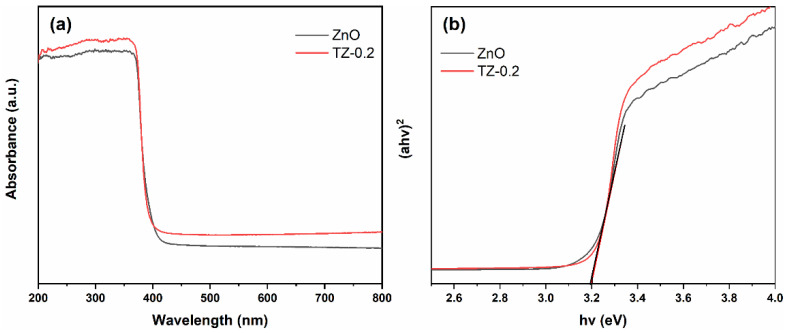
The UV–vis absorption spectra (**a**) and optical band gap of samples (**b**).

**Figure 7 nanomaterials-12-01345-f007:**
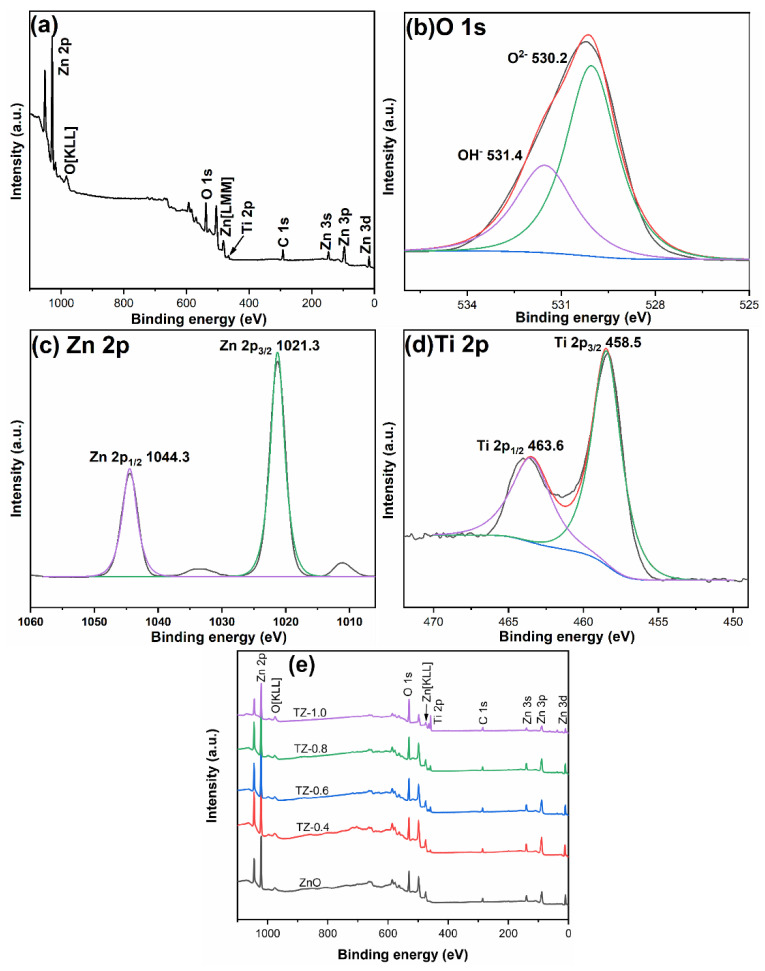
XPS spectra of TZ-0.2 (**a**–**d**) and XPS survey spectra of other samples (**e**).

**Figure 8 nanomaterials-12-01345-f008:**
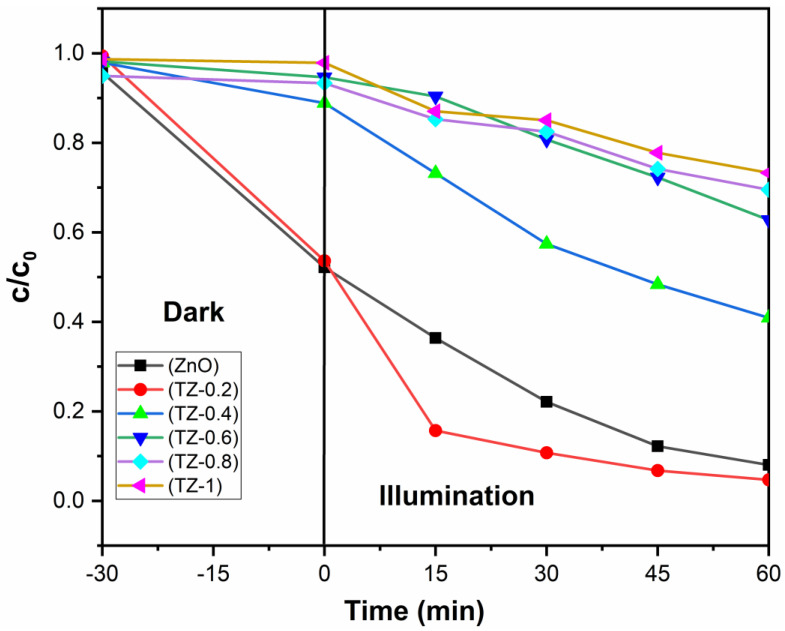
Photocatalytic degradation degree curves of RhB.

**Figure 9 nanomaterials-12-01345-f009:**
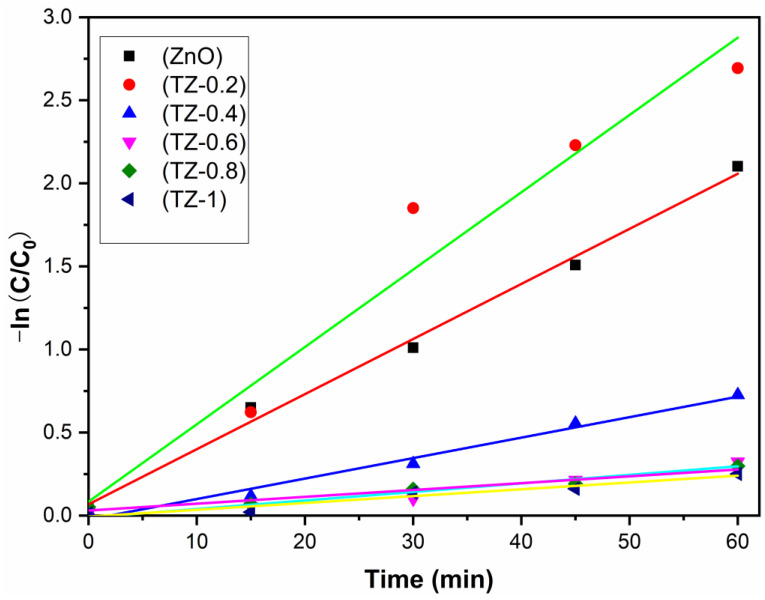
Photodegradation kinetics curves of RhB.

**Table 1 nanomaterials-12-01345-t001:** Binding energies of the samples.

	Zn		O		Ti	
Sample	2p_1/2_	2p_3/2_	O^2−^	OH^−^	2p_1/2_	2p_3/2_
ZnO	1044.9	1021.8	530.2	531.8		
Ti-0.2	1044.3	1021.3	530.2	531.4	463.6	458.5
Ti-0.4	1044.5	1021.4	530.1	531.9	463.9	458.5
Ti-0.6	1044.5	1021.4	530.0	531.6	464.0	458.4
Ti-0.8	1044.5	1021.4	529.9	531.0	463.1	458.3
Ti-1.0	1044.3	1021.2	529.6	530.9	463.8	458.0

**Table 2 nanomaterials-12-01345-t002:** Element compositions of the samples.

Sample	O (at%)	Ti (at%)	Zn (at%)	[Ti]/[Zn]	[O]/([Ti] + [Zn])
pure ZnO	53.78	0	46.22	0	1.16
TZ-0.2	55.42	5.64	38.94	0.14	1.24
TZ-0.4	58.47	9.46	32.07	0.29	1.40
TZ-0.6	49.20	13.37	37.43	0.36	0.97
TZ-0.8	57.36	14.14	28.50	0.50	1.34
TZ-1	54.38	20.45	25.17	0.81	1.19

## Data Availability

Data is contained within the article.

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
