# Peer review of "Preparation, Characterization of ZnTiO3/ZnO Composite Materials and Their Photocatalytic Performance"

_nanomaterials, 2022, doi:10.3390/nano12081345_

Round 1

Reviewer 1 Report

The manuscript titled “Preparation, Characterization of ZnTiO3/ZnO Composite Materials and Their Photocatalytic Performance” presented by Mao Tang, Shiji Lu, Lili He, Xiaodong Zhu, Wei Feng, and Wanming Zhang deals with the composite materials based on ZnTiO3/ZnO that could be used for sewage purification. However, there are issues that could be clarified or took into account:

  • The abstract should not content the abbreviations, for example, XRD, SEM, TEM, etc.
  • The molar ratio does not be measured in percent.
  • Line 17: Capital letter at “When”.
  • Line 22: amorphous content – what the authors want to say.
  • The abstract should be rewritten. In the current form it is not represent the study.
  • Section 2.2. The absence of description of used setups. The advice for authors, please, see how it is presented by the other scientists.
  • Line 76: Not spectrum, but XRD pattern!
  • Line 76-78:” The diffraction peaks in the spectrum of pure ZnO appearing at 31.8°, 34.4°, 36.3°, 47.6°, 56.6°, 62.9°, 66.4°, 67.9° are indexed to the ZnO hexagonal phase (100), (002), (101), (102), (110), (103), (200), (112) crystal planes, indicating that ZnO shows a hexagonal wurtzite structure” should be corrected. For example, “The diffraction peaks in the XRD pattern of pure ZnO appearing at 31.8°, 34.4°, 36.3°, 47.6°, 56.6°, 62.9°, 66.4°, 67.9° are indexed to the ZnO hexagonal phase, with corresponding reflections (100), (002), (101), (102), (110), (103), (200), (112), indicating that ZnO shows a hexagonal wurtzite structure”.
  • Line 87-89: The absence information for which reflect the Sherrer formula was applied.
  • Line 101: The figures content unacceptable symbols. The labels are unreadable.
  • Section 3.4. What is the PL?
  • Line 150: Please, use the term “UV-vis”.
  • Section 3.5 “Element composition and state” dedicated to XPS analysis looks poorly. First of all I recommend the authors to see how the XPS information is presented by the other scientists. Secondly, the XPS analysis should be carried out for all samples. The [Ti]/[Zn] and [O]/([Ti]+[Zn]) surface ratios as well as binding energies (cation states) should be collected for all sample in the Table.
  • The authors use the term “valence” in the wrong way.

The choice of methods to study the catalysts looks well. The XPS analysis should be done for all catalysts – the surface concentration of cations should be presented. The manuscript demands a strong proofreading.

Author Response

The response to Reviewer 1 is attached.

Reviewer 2 Report

I read the manuscript "Preparation, Characterization of ZnTiO3/ZnO Composite Materials and Their Photocatalytic Performance". I found some problems especially in the XPS part. I do not recommend publication in the present form and would like the authors to consider the following points. 

  1. The abbreviations appear suddenly. The authors should define XRD, SEM, TEM, BET, PL, DRS and XPS, like x-ray photoemission spectroscopy.
  2. The wavelength (or photon energy) used in XRD is not written. Please write.
  3. Figure 2 caption says "SEM images of pure ZnO (a, b) and TZ-0.2 (c, d) and elemental mapping images of TZ-103 0.2." The authors should also mention (e-f).
  4. The authors should show EDX data if they want to discuss element distributions.
  5. What is an excitation energy for PL measurements?
  6. In the XPS figures, the higher binding energy should be on the left side. This seems more common these days. 
  7. What is the photon energy for XPS?
  8. What are the small structures at ~ 1010 and ~ 1032 eV?
  9. In the figures, 2P should be 2p to be consistent with the text. In panel (a), the label of Zn 2p is not clear.
  10. What do the authors mean by "high-resolution spectrum"? The authors should show the values of energy resolutions.

Author Response

The response to Reviewer 2 is attached.

Round 2

Reviewer 1 Report

The current form looks well. The authors did a good work under manuscript and answers all comments. However, there are few issues that should be taken into account:

  1. Line 15: shoud be written - "ratio reaches 0.2", but not 20%. As I noted before, ratio is not measured in percent. Similar corrections should be done for another statements (for example, line 17, 20).
  2. Line 49-50: Brunauer-Emmett-Teller surface area (BET) measurment, photoluminescence spectroscopy.

The manuscript demands minor corrections and could be accepted after corrections without additional reviewing.

Author Response

  1. Line 15: should be written - "ratio reaches 0.2", but not 20%. As I noted before, ratio is not measured in percent. Similar corrections should be done for another statements (for example, line 17, 20).

A: Thanks for the reviewer’s careful review. All the expressions have been corrected and mark in red in the revised manuscript.

  1. Line 49-50: Brunauer-Emmett-Teller surface area (BET) measurement, photoluminescence spectroscopy.

A: Thanks for the reviewer’s careful review. All the expressions have been corrected and mark in red in the revised manuscript.

The manuscript demands minor corrections and could be accepted after corrections without additional reviewing.

A: Thanks for the reviewer’s suggestion and comments, which are of great help to improve the quality of our manuscript.

Reviewer 2 Report

I do not find problems in the present manuscript and would like to recommend publication.

Author Response

 Thanks for the reviewer’s suggestion and comments, which are of great help to improve the quality of our manuscript.